# A Hybrid Material Combined Copper Oxide with Graphene for an Oxygen Reduction Reaction in an Alkaline Medium

**DOI:** 10.3390/molecules24030441

**Published:** 2019-01-26

**Authors:** Jiemei Yu, Taizhong Huang, Zhankun Jiang, Min Sun, Chengchun Tang

**Affiliations:** 1School of Chemistry and Chemical Engineering, University of Jinan, 336 West Nanxinzhuang Road, Jinan 250022, China; chm_yujm@ujn.edu.cn (J.Y.); chm_huangtz@ujn.edu.cn (T.H.); chm_jiangzk@ujn.edu.cn (Z.J.); 2School of Materials Science and Engineering, Hebei University of Technology, 8 of First Road of Dingzigu, Hongqiao District, Tianjin 300130, China

**Keywords:** copper oxide, graphene, electrocatalyst, oxygen reduction reaction, alkaline medium

## Abstract

In this work, an electrode material based on CuO nanoparticles (NPs)/graphene (G) is developed for ORR in alkaline medium. According to the characterization of scanning electron microscope and transmission electron microscope, CuO NPs are uniformly distributed on the wrinkled G sheets. The X-ray diffraction test reveals that the phase of CuO is monoclinic. The CuO/G hybrid electrode exhibits a positive onset potential (0.8 V), high cathodic current density (3.79 × 10^−5^ mA/cm^2^) and high electron transfer number (four-electron from O_2_ to H_2_O) for ORR in alkaline media. Compared with commercial Pt/C electrocatalyst, the CuO/G electrode also shows superior fuel durability. The high electrocatalytic activity and durability are attribute to the strong coupling between CuO NPs and G nanosheets.

## 1. Introduction

Our modern societies have to stand in a transition line from traditional fossil energy to sustainable clean energy, with the increasing demands for environmental and energy sustainability [1,2]. In 1839, the first fuel cell was developed which can transform chemical energy directly into electrical energy [3]. Ever since then, more and more attentions have concentrated on this renewable energy technologies due to the excellent energy conversion and relatively high power density [4]. In the development of fuel cells, a considerable number of studies focused on improving its oxygen reduction reaction (ORR) performance [5,6,7,8]. Triggered by the high cost and low tolerance to fuel crossover of Pt-based cathodes, a large number of researches have been conducted on seeking abundant and cheap substituted catalysts for an efficient ORR under fuel-cell working conditions [9]. Among many catalysts developed over the past decades, various non-noble metal catalysts have been considered [10], these include mixed valence oxides of transition metals, nitrogen containing and functionalized carbon materials [11,12,13].

Among the transition metal catalysts, Cu [14], Co [15], and Fe [16,17] are the widely studied materials, particularly, copper based catalysts have been widely investigated to catalyze the ORR due to its good durability, high catalytic activity and environmentally friendliness in alkaline medium [18,19]. Gewirth and co-worker [20] synthesized a Cu-based molecular catalyst with enhanced electrocatalytic properties for oxygen reduction reaction. Meanwhile, Wang et al. [21] designed the material of Cu-complex-hybrid anchored on reduced graphene oxide (rGO) sheets, which has the high exchanged electron density during ORR process. High electrocatalytic activities of Cu_x_Co_3−x_O_4_ for the ORR have also been reported by Koninck et al. [22,23]. Conchi et al. [24] reported a new Cu/rGO catalyst with the excellent ORR activity in terms of high current densities and long-term stability. Blanford et al. [25] attached Cu-containing multi-Cu oxidases (fungal laccases) to a substrate-like anthracene-modified pyrolytic graphite. It was found that, the strong adsorption of laccase on anthracene-based units make the laccase more stable on the modified graphite than that on the untreated graphite. 

In this study, the catalyst of CuO nanoparticles (NPs) anchored on graphene nanosheets (donated as CuO/G) was synthesized by an efficient and facile hydrothermal method. The graphene oxide was reduced simultaneously with the growth of CuO NPs on graphene oxide. The obtained CuO/G hybrid was used as a new electrode material for ORR. The electrocatalytic activity of CuO/G was evaluated, using CuO and G as references.

## 2. Results and Discussion

### Structural Characterization

The morphology and microstructure of our materials were characterized by SEM (scanning electron microscopy), TEM (transmission electron microscopy), and HRTEM (high resolution TEM). In Figure 1a, CuO NPs aggregated to a certain extent. While in Figure 1b, CuO NPs dispersed on the surface of G uniformly, and G nanosheets serve as the support frame. The catalyst loading per area of CuO/G is calculated to be 0.0084 g/m^2^. TEM and HRTEM images of our samples are shown in Figure 1c–f. As can be seen in Figure 1c that CuO NPs are aggregated into large clusters with an average size of about 10 nm, while in Figure 1e, CuO NPs anchor on G nanosheets homogenously, with a small degree of local aggregation and narrow particle size distribution in the range of 6~8 nm. It reveals that CuO NPs in CuO/G hybrid shows smaller particle size than pure CuO. This indicates the dispersing effect of graphene preventing CuO NPs from aggregation. Figure 1d,f show typical HRTEM images of CuO NPs in CuO and CuO/G, a clear lattice spacing of 0.25 nm can be observed, which matches well with the (111) plane of monoclinic CuO [26].

Figure 2a shows the XRD pattern of our samples. The characteristic peaks at 35.50°, 38.79°, and 61.54° match well with the (002), (111), and (113) plans of monoclinic CuO (PDF: 80-1917), respectively, indicating the completely converting from CuCl_2_·2H_2_O to CuO NPs during the hydrothermal process. Table 1 shows the lattice constants of CuO and CuO/G, which consistent well with the results of the previous literature [27].

Meanwhile, the crystallite sizes of CuO and CuO/rGO can be calculated from Equation (1) (Appendix A). The calculations suggest the crystallite size of CuO and CuO/G as 10.7 nm and 7.1 nm, respectively, consist with the TEM results approximately. The XRD pattern of CuO/G also shows a broad peak at ~25°, corresponding to the (002) plane of graphene, indicating an effective deoxygenation of GO during the hydrothermal process. However, the graphene peak is very weak, attributes probably to the relatively lower diffraction intensity and the low amount of graphene compared to CuO. The existence of G in CuO/G is also proved by Raman test. In the Raman spectrum, the *G* band is assigned to the E_2g_ vibration mode of sp^2^ domain indicative of the degree of graphitization, while *D* band represents the structural defects and disordered structures [28]. As shown in Figure 2b, the intensity ration of *D* to *G* peak (*I_D_*/*I_G_*) for CuO/G is 1.0, the low of *I_D_*/*I_G_* value implies the removal of oxygenated functional groups during the reduction of graphite oxide into graphene. XPS spectras were recorded to verify the chemical states information of ions of CuO and CuO/G. As expected, CuO and CuO/G mainly contain Cu, O and C elements (Figure 2c). The C element of CuO/G results from the adventitious element carbon and the G molecules, while the C element of CuO only results from the contamination of the testing environment. The Cu 2p spectrum of CuO and CuO/G, as shown in Figure 2d, display double peaks. The peak located at 953.1 eV and 932.2 eV are assigned to Cu 2p3/2 and Cu 2p1/2 of Cu^2+^ in CuO, while the corresponding peaks of Cu 2p in CuO/G are observed at 957.5 eV and 936.6 eV, shifting towards the higher energy region by 4.4 eV. Such a shift towards the higher energy region of Cu 2p in CuO/G may be attributed to the strong interaction between CuO and G. This also indicates that the coupling between CuO and graphene occurs through the Cu^2+^ species of CuO [18,29]. Figure 2e shows the O 1s XPS spectrum region of CuO/G catalyst. The peak located at 530.0 eV and 532.1 eV correspond to O^2−^ ions of the Cu-O species and the oxygen-deficient regions within the CuO matrix, respectively. The peak located at 533.7 eV is assigned to the OH group or chemisorbed oxygen on the surface of the catalyst. The presence of the OH or oxygen groups absorbed onto the surface of CuO/G may helpful to enhance its electrocatalytic performance [30]. Figure 2f shows the high-resolution XPS spectrum of C 1s, which can be deconvoluted into three components as C-C (283.9 eV), C-O (286.1 eV) and C=O (287.8 eV) species [31], respectively. The C-C is the predominant specie by measuring the relative peak area, indicating the GO has been well deoxygenated during the hydrothermal procedure [32].

Figure 3a–d show the CVs of G, CuO and CuO/G in Ar and then in O_2_-saturated 0.1 M KOH from 1.2 to 0.2 V at a scan rate of 5 mV/s vs. RHE reference electrode. It can be seen in Figure 3a–c that the CV tests of G, CuO, and CuO/G are essentially featureless under Ar atmosphere in the working potential range. The same phenomenon is also revealed at the G electrode in O_2_-saturated electrolyte, indicating that graphene is inherently electrochemically silent in this condition. As shown in Figure 3b,c, CuO and CuO/G electrodes display a significant oxygen reduction peak in O_2_ atmosphere. According to the Pourbaix Diagrams of Cu-H_2_O system (Appendix A), the dominant species are Cu_2_O and CuO at pH around 13 with the potential rang of 1.2~0.2. While in our experiment conditions, the main specie is CuO. When the electrode was stabilized in 0.1 M KOH solution saturated with Ar, the redox peak due to Cu(I)/Cu(II) should be revealed in the anodic sweep, as indicated in Equation (1). However, there is no obvious redox peaks observed in Figure 3b, which may ascribing to the weak current density of this process that couldn’t be detected under this condition:(1)$${\mathrm{Cu}}_{2}O+2\mathrm{OH}\to 2\mathrm{CuO}+H_{2}O+2e^{-}$$

The ORR activity of the three electrodes increases as follows: G < CuO < CuO/G, as clearly evidenced by the onset potential shown in Figure 3d. The onset potential of CuO/G is comparable to that of Cu_x_Co_3−x_O_4_ [22,23] and Cu/rGO [24]. The results above demonstrate that graphene can strongly boost the electrocatalytic activity of CuO NPs although it has no catalytic activity alone [26,33]. However, the peak current intensities of CuO/G is lower than that of Cu_x_Co_3−x_O_4_, Cu/rGO, and commercial Pt/C catalyst [34], which should be attributed to the slightly aggregation of the CuO NPs that decreased the numbers of activity center for ORR. The diffusion-current-corrected Tafel plot of specific ORR activity of CuO and CuO/G are show in Figure 3e. The Tafel slope are 207 and 141 mv/dec for CuO and CuO/G, respectively. The Tafel slope of CuO/G is smaller than that of CuO, the lower Tafel slope indicates the higher intrinsic catalytic activity [29]. To construct the Tafel plots, the exchange current density is derived from the mass-transport correction using Equation (2) (Appendix A).

As shown in Table 2, the exchange current density of ORR is 2.12 × 10^−8^ for CuO and 3.79 × 10^−5^ mA/cm^2^ for CuO/G, and the value of CuO/G is approximate to the commercial Pt/C which the ORR exchange current density is about 10^−6^–10^−9^ mA cm^−2^ [34]. It also clearly shows that the *i*_0_ of CuO/G is higher than that of CuO, which can be explained by the synergistic contributions between CuO NPs and G sheets. It is clearly seen in Figure 1b,e that CuO nanoparticles are densely anchored on the G single layer which is very important, because this ensures the efficient electron collection via the rGO sheets during the ORR processes. Figure 3f shows the EIS of CuO and CuO/G. It is obvious that the resistance of CuO/G electrode is smaller than that of CuO, owing to the conductive G nanosheets improve the electron transfer in CuO/G materials. The fitted equivalent circuit of CuO/G is shown in Figure 3g, the obtained Rs1, Rs2, Rp1 and Rp2 are 1128, 100, 1303, and 6571 Ω, respectively. Figure 3h shows the long-term stability of the CuO and CuO/G electrode, by running the ORR for 16,000 s at a fixed potential. The test of commercial Pt/C is also evaluated for comparison. The data collected suggests a high stability of CuO and CuO/G with a slight performance attenuation after 16,000 s of operation, while commercial Pt/C shows a gradual degradation. This confirms the strong interactions between the CuO NPs and the graphene, which preventing active sites from losing during cycling [24].

Figure 4a shows the polarization curves of CuO nanohybrid at different rotation rates. All the curves reach well-defined diffusion limiting currents. In order to quantitatively understand the ORR activity of CuO electrode, the *K-L* plots at different rotating rates are analyzed (Figure 4b). As can be seen, all the *K-L* plots display good linearity. According to *K-L* equations (Equations (3) and (4)) (Appendix A), the electron transfer number (n) for per oxygen molecule can be estimated from the intercepts and slopes. The result reveals that the n of CuO calculated from the B-factor is ranges from 3.18 to 3.35 [35]. Similar profiles can be also seen from the CuO/G catalyst, as shown in Figure 4c,d, the calculated electron transfer number is in the range of 3.84~3.92, indicating a four-electron mechanistic pathway, which is similar to the previous studies of Gewirth and Wang et al. [21,36]. However, the ORR catalyzed by individual copper oxide behave in a mixed mechanistic fashion operating with two- and four-electrons [37]. As has been reported that depending on the metal-graphene spacing and the spinel structure of transition metal oxides, charge transfer may exist between the transition metal oxides and graphene, which may be the real reason that certain transition metal oxides NPs supported on graphene surface exhibit enhanced catalytic activities [38].

According to the Pourbaix Diagrams of O_2_-H_2_O system, the ORR can happen through a four-electron or two-electron process at pH around 13 with the potential rang of 1.2~0.2:(2)$$O_{2}+2H_{2}O+4e^{-}\to 4{\mathrm{OH}}^{-}$$
(3)$$O_{2}+2H_{2}O+2e^{-}\to 2{\mathrm{OH}}^{-}+H_{2}O_{2}$$
(4)$$H_{2}O_{2}+2e^{-}\to 2{\mathrm{OH}}^{-}$$

In order to further probe the ORR mechanism taking place on the two electrodes simply and exactly, rotating ring-disk electrode (RRDE) were performed to monitor the electron transfer number and the formation of hydrogen peroxide (H_2_O_2_) during the ORR process, in O_2_-saturated 0.1 M KOH solution with a scan rate of 5 mV/s at 1600 rpm. As shown in Figure 4e,g, CuO/G exhibits higher disk current and lower ring current (peroxide oxidation), compared with CuO catalyst. Based on rotating ring-disk voltammograms, the electron transfer number and the current efficiency for hydrogen peroxide formation (H_2_O_2_ %) during the ORR process are determined quantitatively according to Equation (5) and (6) (Appendix A). As can be seen in Figure 4f, the measured H_2_O_2_ yield of CuO catalyst is lower than 14%. Specifically, the average transferred electron number is found to vary between 3.1 and 3.4, indicating that ORR at the CuO electrode proceeds via both two-electron and four-electron pathways. While, seen in Figure 4h, the H_2_O_2_ yield of CuO/G is less than 2.5%, accordingly, the electron transfer number is found to vary between 3.8 and 4.0. These results suggest that the ORR catalyzed by CuO/G is mainly through a four-electron pathway by directly forming hydroxyl species as the final products.

## 3. Experimental

### Sample Preparation

Graphene oxide (GO) sheets were prepared by a modified Hummers method in our laboratory [36,39]. The CuO/G is synthesized by a one pot solvothermal method as illustrated in Scheme 1. 1.70 g copper chloride (CuCl_2_·2H_2_O) was first dispersed into 40 mL deionized water, followed by the addition of 6 mL GO (6 g/L) and 1.60 g sodium hydroxide. After magnetic stirring for 10 min, 50 mL absolute ethyl alcohol and 2 mL ethylene glycol were added into the solution. The obtained solution was removed to a teflon centrality that sealed by stainless cylinder and kept at 180 °C for 24 h. After naturally cooling down, the precipitates were collected, and thoroughly washed with ethanol and deionized water. Finally, the obtained powder was dried at 60–80 °C in vacuum for characterization. Then 0.79 g CuO/G was obtained, and the yield of reaction was calculated to be about 95%. Moreover, the samples without CuO or G were also synthesized for comparison under the same condition, and were denoted as G and CuO.

## 4. Conclusions

We have demonstrated a facile route to fabricate CuO/G hybrid. Due to the enhanced kinetics for electron transfer resulting from its unique structure of CuO NPs dispersed on conductive graphene layers, the CuO/G electrode exhibits excellent electrocatalytic performance, in terms of high cathodic current density, positive onset potential, and a high electron transfer number for ORR in 0.1 M KOH solution. The electron transfer numbers for the ORR on the CuO/G electrode is calculated to be about 3.9, indicating a four-electron mechanistic pathway from O_2_ to OH^−^. In addition, the CuO/G electrode also shows superior fuel durability compared to the commercial Pt/C catalyst after a long-term operation of 16,000 s. Thus, it is expected that the CuO/G hybrid could have a great potential as a non-precious metal-based catalyst towards the ORR.

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
