# Peer review of "A Hybrid Material Combined Copper Oxide with Graphene for an Oxygen Reduction Reaction in an Alkaline Medium"

_molecules, 2019, doi:10.3390/molecules24030441_

Round 1

Reviewer 1 Report

In this modified version, the authors have compared their results with previous results, and explained why CuO/G has lower peak current intensities than those of CuxCo3-xO4 and Cu/rGO. A technique of obtaining wrinkled graphene sheets has also been explained, and the influences of wrinkled graphene have been illustrated. The format of this manuscript has been revised properly. The supplementary material is helpful to comprehensively understand this manuscript. This manuscript should be good to publish.

This modified version has explained my previous questions, the supplementary material is detailed and helpful. My recommendation is this manuscript is good to publish.

Author Response

Dear reviewer,

Thank you for your help and kindness during the review process.

Wish you a good day.

Yours sincerely

Min Sun

Reviewer 2 Report

In the manuscript submitted, the authors describe the formation of CuO/Gr sheets and the application in ORR.  Although they present the mechanistic study in detail, a key concern is the resulting structural integrity.  (1) How strong is the binding between CuO and Gr?  This pertains to the stability during ORR.  (2) How uniform is the decoration of CuO NPs on the surfaces of Gr?  This is also matter of reproducibility of ORR measurements.  In this aspect, the authors should provide the experimental evidence in order to show the structural integrity of resulting structures.  If the authors meet the above requirements, this reviewer would recommend it for publication after mandatory revision.

Author Response

Dear reviewer,

Thank you for your help during the review process. We have carefully read through the comments from you and would like to thank you for your very thoughtful comments and suggestions. We have accordingly revised the manuscript (highlighted in RED) to address your comments. Below are our responses (marked as Authors’ Response following each comment from you) to your comments.

Reviewer:

In the manuscript submitted, the authors describe the formation of CuO/Gr sheets and the application in ORR. Although they present the mechanistic study in detail, a key concern is the resulting structural integrity. 

Reviewer: (1) How strong is the binding between CuO and Gr?  This pertains to the stability during ORR. 

Authors’ Response: Dear reviewer, Thank you for your insight comments. We have added the strong binding effect between CuO and Gr.

Page 4 line 108:

…while the corresponding peaks of Cu 2p in CuO/G are observed at 957.5 eV and 936.6 eV, shifting towards the higher energy region by 4.4 eV. Such a shift towards the higher energy region of Cu 2p in CuO/G may be attributed to the strong interaction between CuO and G. This also indicates that the coupling between CuO and graphene occurs through the Cu2+ species of CuO [18, 30].

Page 7 line 168:

This confirms the strong interactions between the CuO NPs and the graphene, which preventing active sites from losing during cycling [24].

Reviewer: (2) How uniform is the decoration of CuO NPs on the surfaces of Gr? This is also matter of reproducibility of ORR measurements. In this aspect, the authors should provide the experimental evidence in order to show the structural integrity of resulting structures. If the authors meet the above requirements, this reviewer would recommend it for publication after mandatory revision.

Authors’ Response: Dear reviewer, thank you for your comments and excellent suggestions. We have describe the uniform extent of CuO NPs on the surfaces of Gr. In order to explain it more clearly, we also added the catalyst loading per area in our manuscript.

Page 2 line 73:

While in Fig. 1b, CuO NPs dispersed on the surface of G uniformly, and G nanosheets serve as the support frame. The catalyst loading per area of CuO/G is calculated to be 0.0084 g/m2.

Page 2 line 75:

As can be seen in Fig. 1c that CuO NPs are aggregated into large clusters with an average size of about 10 nm, while in Fig.1e, CuO NPs anchor on G nanosheets homogenously, with a small degree of local aggregation and narrow particle size distribution in the range of 6~8 nm.

Round 2

Reviewer 2 Report

It seems that the authors met the requests by the reviewer.  I would like to recommend it for publication in Molecules as is.